



**Mantle roots of the Emeishan plume: an evaluation based on telesismic P-wave**
**tomography**
Chuansong He[1]*, M. Santosh[2, 3]
[1]*Institute of Geophysics, China Earthquake Administration, Beijing 100081, China*
[2]*Centre for Tectonics, Resources and Exploration, Department of Earth Sciences,*
*University of Adelaide, SA 5005, Australia*
[3]*School of Earth Sciences and Resources, China University of Geosciences*
*Beijing, 29 Xueyuan Road, Beijing 100083, China*
Abstract: The voluminous magmatism associated with Large Igneous Provinces (LIP)
is commonly correlated to upwelling plumes from the Mantle Transition Zone or the Core-
Mantle Boundary (CMB). Here we analyse seismic tomographic data from the Emeishan
LIP in southwestern China. Our results reveal vestiges of delaminated crustal and (or)
lithospheric material in the cental part of the study area, and upwelling mantle in the
southern part. Our results do not provide any conclusive evidence for upwelling mantle
plume rooted in the CMB beneath the Emeishan LIP. We therefore suggest that the
magmatism and the Emeishan LIP formation might be connected with the melting of
delaminated lower crustal and (or) lithospheric components and associated plume-like
upwelling from the mantle transition zone.
Key word: Emeishan Large Igneous Province; Teleseismic P-wave tomography;

* Corresponding author. Tel.: 86-10-68729303; Fax: 86-10-83534760.
*Email address*: hechuansong@aliyun.com



Lithospheric and (or) delamination; Mantle plume; Tectonics.
**1. Introduction**
The large-scale and transient magmatic events on the globe at different times during
Earth history are closely linked to mantle dynamics (Coffin and Eldholm, 2001; Ernst and
Buchan, 2001). The punctuated but intense magmatic activities over the globe has
generated several Large Igneous Provinces (LIPs) in different regions (Uenzelmann-
Neben, 2013; Pirajno and Hoatson, 2012). Mantle plumes which are upwellings of hot
material from deeper parts of the Earth (Arndt, 2000) have been invoked to explain the
link between LIP and modern volcanoes. LIPs are characterized by large lava
outpourings, such as those found in Siberia, India, and Emeishan, which also have
important implications in surface environmental changes including mass extinctions
(Buiter, 2014; Wignall, 2011).

Mantle upwellings received attention when Wilson (1963) suggested that the
Hawaiian Islands were produced when oceanic lithosphere moved over a stationary 'hot
spot' in the mantle, following which the role of plumes and their relation to mantle
convection was further realized (Morgan, 1971). It is now widely recognized that
upwelling mantle plumes generate many LIPs and numerous small chains of seamounts
(Griffiths and Campbell, 1990; Hofmann, 1997; Maruyama et al., 2007; Safonova et al.,
2009; White, 2010; Dobretsov, 2011, Safonova and Santosh, 2014). When upwelling
mantle plume impinges on continental or oceanic lithosphere, large-scale eruption and





intrusion of mafic and ultramafic melts occur generating LIPs (Coffin and Eldhom, 1992,
Pirajno, 2007; Sheth, 2007b; Bryan and Ernst, 2008; Shellnutt and Iizuka, 2012).

The basaltic rocks of LIPs have been investigated to understand the source, nature

and tectonic setting of LIPs formation (Smith and Asimow, 2005; Herzberg and Asimow,
2008; Shellnutt and Iizuka, 2012). The Emeishan basalts (ca. 257 - 262 Ma) in southwest
China are exposed over an area of 0.25-0.3 million square kilometers in the Sichuan,
Yunnan and Guizhou provinces comprising a total volume of about 0.25 million km$^3$
(Huang and Opdyke, 1998), with thickness of the basaltic flow ranging from one to two
hundred meters in the eastern part to more than five kilometers in the west (Ali et al.,
2010; Deng et al., 2010). The Emeishan LIP hosts some of the world-class Fe–Ti–V
oxide and Ni–Cu sulfide deposits (Pang et al., 2013; Zhou et al., 2013). The region has
been divided into three zones (inner, intermediate and outer) (Fig. 1) based on
biostratigraphic, sedimentological and geochemical characteristics (Xu et al., 2001; Deng
et al., 2010).

Previous studies suggested the Emeishan flood basalts with mantle plume

impingement at the base of the lithosphere causing large-scale regional up-doming prior
to volcanism (Shellnutt et al., 2012; Shellnutt, 2013, 2014; Li et al., 2002; Gao et al.,
1999; Liu et al., 2008) with a short eruption period of less than 1 Ma (Song et al. 2004).
However, the primary evidence for upwelling mantle plume has remained elusive. Some
workers (e.g. Ukstins Peate and Bryan, 2008) have also challenged the concept of



upwelling mantle plume leading to LIP formation in the Emeishan area. It has also been
argued that that submarine volcanism took place during emplacement of the Emeishan
LIP and that some lava flows close to the centre of the LIP were erupted in a submarine
setting (Ukstins Peate and Bryan, 2008; Ali et al., 2010). This model considers that the
products of initial eruption were extruded at or around sea level, and that the moderately
positive topography is a reflection of the rapid accumulation of the volcanic pile (Peate
and Bryan, 2008). Therefore, in order to further understand the Emeishan LIP formation,
it is necessary to investigate the deep structure or mantle dynamics beneath the
Eemishan LIP area.

In the past two decades, seismic tomography has increasingly found application
as a potential tool to explore the heterogeneous structure of the Earth's interior, which in
turn is important to gain insights into mantle dynamics and crust-mantle interaction
processes. Several seismic tomographic studies have been carried out on the Emeishan
LIP and surrounding regions, including 2.5 dimensional tomography of the uppermost
mantle (Lü et al., 2014), ambient noise Love and Rayleigh wave tomography (Li et al.,
2010, 2009), teleseismic P-wave tomography (Yang et al., 2014; Huang et al., 2015; Bai
et al., 2011), local earthquake tomography (Huang et al., 2009; Xu et al., 2012),
interstation Pg and Sg differential traveltime tomography (Li et al., 2014) and Pn
anisotropic tomography (Lei et al., 2014). Generally, most of these studies targeted the
crustal and upper mantle velocity structure in this area.



In this study, we carried out a systematic tomographic analyses with a view to
construct the velocity structure or mantle dynamics of the upper mantle beneath the
Emeishan LIP area. The results provide a vivid image of the upwelling mantle and the
lower crustal and (or) lithospheric delamination. Our results further demonstrate the
dynamic relationship between delamination and upwelling mantle.

**2.  Data and method**


The basic principle for teleseismic tomography assumes that the relative travel-time
residuals resulted from the heterogeneity in the model space (e.g. Yang et al., 2014;
Zhao et al., 1992). The location of the seismic ray crosses through the boundary of the
study region was determined by a 1D velocity model and theoretical travel time and
seismic ray paths are obtained by the fast raytracing technique (Zhao et al., 1992, Yang
et al., 2014). 3D grids are employed to express the velocity perturbation values, and any
point in the model space can be calculated from values of the surrounding eight nodes by
linear interpolation (Zhao et al., 1992, 1994, 1997, 2009; 2002, 2013; Zhao and Lei,
2004; Zhao and Ohtani, 2009;, Zhao, 2001, 2004).

In this study, we collected data recorded by China seismic network from July 2007 to
March 2014 which comprises 228 seismic stations in the study region (Fig. 1, Fig. 2). The
371 seismic events were selected with epicentral distance ranging from 30°-
85°correspond to earthquake magnitude >6.0. *P* arrivals were correlated on the vertical
component after bandpass filtering between 0.3 and 3 Hz. Our assembled data set





contains 42500 *P*-wave arrivals. Based on the distribution of the relative arrival time, we
limited the relative arrival time of >-2s and <2s used to tomographic inversion (Fig. 3). To
analysize this data set, we used the tomographic method of Zhao et al (1994). The three-
dimension grid nodes was set up. The lateral grid spacing is 1° ×1° and the vertical grid
spacing are 50, 100, 200, 300, 400, 500, 600, 700 and 800 km respectively. After the
crustal correction to remove the effect of lateral crustal heterogeneity (crustal correction
depth: 50 km) (Jiang et al., 2015), the velocity perturbations from the one-dimensional
iasp91 Earth model (Kennett and Engdahl, 1991) at each grid node was taken as
unknown parameter. The LSQR algorithm (Paige and Saunders, 1982) was used to solve
the large and sparse system of observation equations with damping and smoothing
regularizations (Zhao, 2004). The optimal value of the damping is based on the trade-off
curve between the RMS travel-time residuals and the norm of the model, after many
tests, and eventually, 15 were adopted as damping parameter (Fig. 4).

The results from tomographic inversion should be assessed along with resolution and
error analyses. The procedure to evaluate the resolution of a tomographic result is to first
calculate a set of travel time delays which result from tracing the actual rays through a
synthetic test structure, followed by the inversion of the delays as though they are data,
and finally comparing the synthetic inversion result with the initial structure (Zhao et al.,
1992). Following this method, the synthesized data were inverted to evalute whether the
assigned checkerboard pattern could be recovered or not. Here, we designed the lateral
grid spacing as 1° ×1° and the vertical grid spacings are 50, 100, 200, 300,   400,   500



600, 700 and 800 km. Positive and negative velocity perturbations of 5% were assigned
to all the 3-D grid nodes. The results show that the resolution is generally high in most
parts of the study area (Fig. 5), except for the marginal region and 50 and 800 km depth
sections. We also carried out the checkboard test along west-east profiles (Fig. 1, Fig.
6 ), all results show high resolution at all profile and the synthetic data can be recovered
at main part, except for western part of the section, the north-south profiles also show
high resolution at most part, except for the marginal region (Fig. 7). The results of the
checkboard test demonstrated our data and calculation adequately meet with the
required resolution for this study.

**3.   Results**

The results from this study show large-scale high velocity perturbation at 50,100,

200, and 300 km depth sections in the northeastern part of the study area or Yangtze
block which reflects the lithospheric root of the Sichuan Basin (Fig. 8). This result is
consistent with previous teleseismic P-wave tomographic studies (Yang et al., 2014;
Huang et al., 2015; Li et al., 2006; Bao et al., 2009). A recent receiver function study
indicates large-scale delamination at the central and southern parts of this area (He et
al., 2014), which might have triggered large-scale mantle convection leading to the high
velocity domain in this region. Therefore, tomographic images show a large-scale high
velocity perturbation at 300 and 400km depth at the central part (Fig. 8, Hv1) which may
be associated with the crustal and (or) lithospheric delamination. Huang et al. (2015) also
defined a large-scale high velocity perturbation at 350 and 400 km depth, which is



consistent with our results. High velocity perturbations are also revealed at 500, 600 and
700 km depth sections (most in the mantle transition zone) (Fig. 8, Hv2), which might
connect with the crustal and (or) lithospheric delamination or vestiges of subduction slab
of the Indian plate (Yang et al., 2014). In the southern part of the study area, large-scale
low velocity perturbations are seen at 50, 100, 200 and 300 km depth section (Fig. 8
Lv1). Furthermore, these low velocity perturbations broadly overlap at different depths,
possibly indicating a connection with the upwelling mantle. Huang et al. (2015) and Yang
et al. (2014) also defined a low velocity perturbation at 100-200 km depth, which is also
consistent with our results. In the 700 and 800 km depth sections, there is an obvious low
velocity perturbation in the southern part of the region, which might represent the vestige
of the upwelling mantle.

In the west-east direction profile (Fig. 9), the high velocity perturbation at the root of

Sichuan basin can be clearly seen in Figs. 9a and b. The large-scale high velocity occurs
in the upper mantle region around 26-28°, which is consistent with the results from 300
and 400 km depth sections (Fig. 8), which further suggests large-scale delamination
process beneath the Emeishan LIP area. Several discontinuous high velocity
perturbations are seen in the mantle transition zone, possibly related to crustal and (or)
lithospheric material delaminated into the mantle transition zone. Alternately, these
features might also correspond to the vestiges of the subduction slab of the Indian plate.
In Fig. 9d, there is a large-scale low velocity perturbation in the upper mantle (along the
24°N), which reflect upwelling mantle originating from the mantle transition zone. Yang et



al. (2014) and Huang et al. (2015) also defined a low velocity perturbation or upwelling
mantle almost at the same location (along the 25°N).

In order to further evaluate our results, we took 4 profiles along the north-south

direction (Fig. 10). In Fig. 10e, there is a low velocity perturbation or large-scale upwelling
mantle, which is consistent with Lv1. In Fig 10f and h, there is a low velocity perturbation
or upwelling mantle originating from the mantle transition zone. In Fig. 10g, the upwelling
mantle may originate from the mantle transition zone, because the low velocity
perturbation is very weak at the lower mantle part, which might also be due to the low
resolution of the profile (Fig. 7).

The tomographic image identified by this study shows an obvious low velocity

perturbation in the upper mantle beneath the southern part of the study area, there are
no vestiges of any upwelling mantle plume beneath the Emeishan LIP. In contrast, there
are the low velocity perturbation in the upper mantle and mantle transition zone, we
speculate that the low velocity perturbation in the southern part of the region (Fig. 8, Fig.
9 and Fig. 10) might be associated with crustal and (or) lithospheric delamination. These
vestiges are also identified within high resolution checkboard test (please see Fig. 5, Fig.
6 and Fig. 7), confirming that our results are reliable.

**4. Discussion**



### 4.1 The location of the Emeishan LIP formation



The south China block docked with the Indochina Block on the southwest in the
Triassic along the Ailaoshan-Song Ma suture, on the west along the Longmenshan Fault,
and on the north with the North China Craton along the Qinling–Tongbai–Hong'an–Dabie–
Sulu orogenic belt (Li et al., 2002; Zhou and Zhu, 1993; Mao et al., 2013; Zheng et al.,
2013). The Emeishan LIP is considered to have formed in the Permian-Triassic (Song et
al., 2013; Chung and Jahn, 1995), suggesting a close link with the tectonics associated
with the block amalgamation. The LIP was broken up by the Red River Fault (Xiao et al.,
2004) and is now bounded by the Longmenshan fault (He et al., 2007). However, the ~260-
Ma Emeishan LIP in SW China and northern Vietnam includes voluminous continental
flood basalts that are believed to have formed from same upwelling mantle (Chung and
Jahn, 1995; Xu et al., 2004; Zhou et al., 2006; Wang et al., 2007). Recent studies also
suggested that tectonic lenses of the same basaltic sequence (Camthuy Formation) and
associated rocks are present in northern Vietnam (Tien, 2000; Shi and Shen, 1998), and
were displaced several hundred kilometers to the southeast by Oligo-Miocene sinistral
motion along the Ailao Shan-Red River Fault (Ali et al., 2005), suggesting that the
Emeishan LIP was formed after the closure of the south China block and the Indochina
Block. Although the paleogeographic location of the region of the LIP is near the equator
in the early Permian (Ali et al., 2005; Enkin et al., 1992), the Emeishan terrane arrived in
the present location in the later Permian or early Triassic prior to the LIP formation.

Recent receiver function study also demonstrated a convective circulation system



between the lower crust and the upper mantle transition zone beneath the Emeishan area
associated with the Emeishan LIP formation (He et al., 2014), which further suggests the
formation of the Emeishan LIP at the present location.
**4.2    The mechanism of the Emeishan LIP formation**

Predictions based on numerical and fluid dynamic modelling show that mantle
plumes originating from either the MTZ or the CMB would result in broad domal uplift
(>1,000km wide, 500 to >1,000m high) preceding volcanism in LIPs (Peate and Bryan,
2008; Campbell and Griffiths, 1990; Richards et al., 1989). However, the location and
distribution of the voluminous mafic volcaniclastic deposits, pillow lavas and marine
sediments in the Emeishan LIP do not confirm with the zonal definition of a broad uplifted
dome (Peate and Bryan, 2008). Therefore, the relationship between dynamic uplift and
plume-related process in the Emeishan LIP has remained equivocal (Peate and Bryan,
2008; Sheth, 2007a; Ali et al., 2010; Shellnutt, 2014).

The rise and impingement of mantle plumes on continental and oceanic lithospheric
plates would lead to the formation of mafic/ultramafic lower crust (Pirajno, 2007).
Although, some of the previous studies indicated a high velocity lower crust beneath the
Emeishan LIP (Xu et al., 2007), suggesting mafic/ultramafic lower crust generated by
lower crustal underplating or the upwelling mantle plume during later Permian (Shellnutt,
2014; Zhong et al., 2009; Xu et al., 2004; Tang et al., 2015; Usuki et al., 2015). However,
the dominantly felsic to intermediate lower crust in this area identified from receiver
function analyses (He et al., 2014, 2009) do not favour any large-scale underplating in



the Emeishan LIP area (He et al., 2014; S.S. Sun et al., 2012).

Alternate models consider that the LIP magmatism was triggered by decompression-
induced melting of upper mantle beneath zones of lithospheric extension or fractures
(Uenzelmann-Neben, 2013) which does not require any upwelling mantle plume. Pre-
eruptive subsidence and asthenospheric flow into voids created by delamination of dense
eclogitic lower crust and (or) lithosphere have been proposed by some workers (Anderson,
2007; Hales et al., 2005), such as in the case of the Siberian trap basalts (Elkins-Tanton
and Hager, 2000).

Tomography studies have indicated a high velocity perturbation zone at 500 and 600 km
depth section identified by this study and the earlier studies in the Emeishan LIP area
(Ferris et al., 2003; Yang et al., 2014). This might link the cold material detached or
delaminated from the lower crust and (or) lithosphere into the upper mantle leading to the
velocity increase.

The crustal and (or) lithospheric delamination can generate mantle upwelling and
extensive volcanism (Vlaar et al., 1994; van Thienen et al., 2004), the scale and extent of
which are related to the intensity of the delamination process. A large-scale lower crustal
and (or) lithospheric delamination or sinking may get arrested at the 660 km discontinuity
identified by this study, where crustal and lithospheric components would be melted
(Lustrino, 2005) because the mantle transition zone (MTZ) is a potential water reservoir

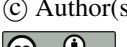



in the Earth's interior (Karato, 2011; Kuritani et al., 2011). Accumulation of subducted
crustal debris, and delaminated crust and (or) lithosphere at the MTZ are speculated to
give rise to 'second continents' on the bottom of the upper mantle (Kawai et al., 2013;
Korenaga, 2004, Lustrino, 2005). The minerals in Earth's mantle transition zone as 'water
tanks' might trigger dehydration melting of vertically flowing mantle (Schmandt et al.,
2014). Because of their buoyancy, crustal and (or) lithospheric melts rise up as plume-
like upwelling instead of being dragged down to the convecting lower mantle (Lustrino,
2005). Thus, lower crustal delamination and mantle inflow are considered to set the ideal
scene for plume-like upwelling from the MTZ (He et al., 2014). Eventually, the plume-like
upwelling resulted in the Emeishan LIP formation.

Meanwhile, removal of the lower crust and (or) lithosphere allows mantle to rise to

shallower depths leading to decompression melting reflected as low velocity
perturbations (Schott and Schmeling, 1998; Elkins-Tanton and Hager, 2000; Elkins-
Tanton, 2005). Accordingly, some low velocity perturbations identified by this study may
be the vestige of the mantle upwelling.

**Conclusions**

The tectonic framework of Emeishan LIP is characterized by the Longmenshan

thrust fault in the northwest and the Ailaoshan-Red River strike slip fault in the southwest.
It is possible that the assembly of Yangtze block with another crustal block in the Late
Permian and Early Triassic might have led to crustal thickening and large-scale


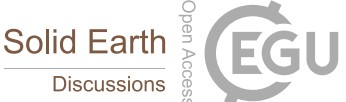

delamination of the lower crust and (or) lithosphere. The delamination resulted in the
upwelling asthenosphere and generation of crustal melts that triggered plume-like
upwelling and Emeishan LIP formation, with no evidence for any large plume rising from
the CMB beneath the Emeishan LIP.

**Acknowledgements**

Waveform data for this study are provided by Data Management Centre of China
National    Seismic    Network    at    Institute    of    Geophysics    (SEISDMC,
doi:10.11998/SeisDmc/SN), China Earthquake Networks Center and CQ, GX, GZ, QH, SC,
XZ, YN Seismic Networks, China Earthquake Administration (Zheng et al., 2010). This
study also contributes to the Foreign Expert funding from China University of Geosciences
Beijing, and Professorial support from University of Adelaide to M. Santosh.

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



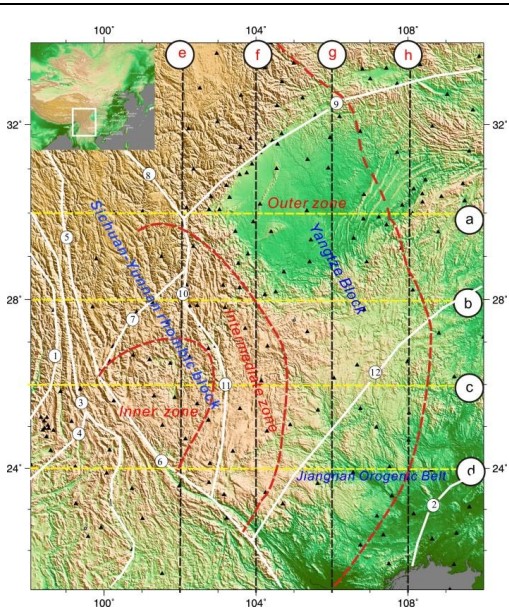


Fig. 1. Tectonic framework, distribution of seismic stations (black triangle) and the west-

east and north-south direction profiles in the Emeishan LIP area. 1: Nujiang fault, 2:

Shaoxing-Jiangshan-Pingxiang fault, 3: Langcangjiang fault, 4: Nandinghe fault, 5: Weixi-

Qiaohou fault, 6: Honghe fault, 7: Yangjiang-Xiaojinhe fault, 8: Xianshuihe fault, 9:

Longmenshan fault, 10: Anninghe-Zhemuhe fault, 11: Xiaojiang fault, 12: Jiujiang-Shitai

buried fault, black triangle: seismic station.

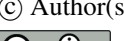



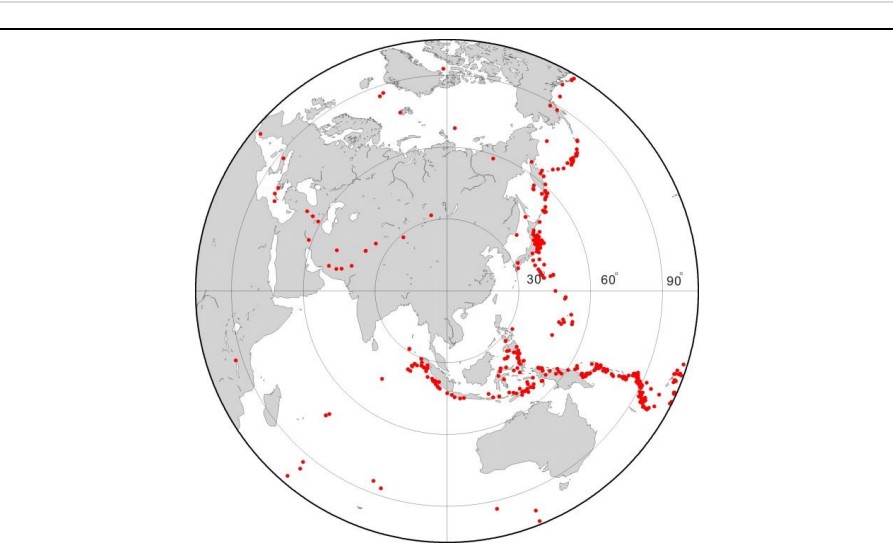


Fig. 2. Seismic events used in this tomographic study. The 371 events with epicenter
distance range from 30 ° to 85 ° for each station-event pair.


Fig. 3 Distribution of relative arrival time. We limited to >-2s and <2s for the tomographic

inversion.





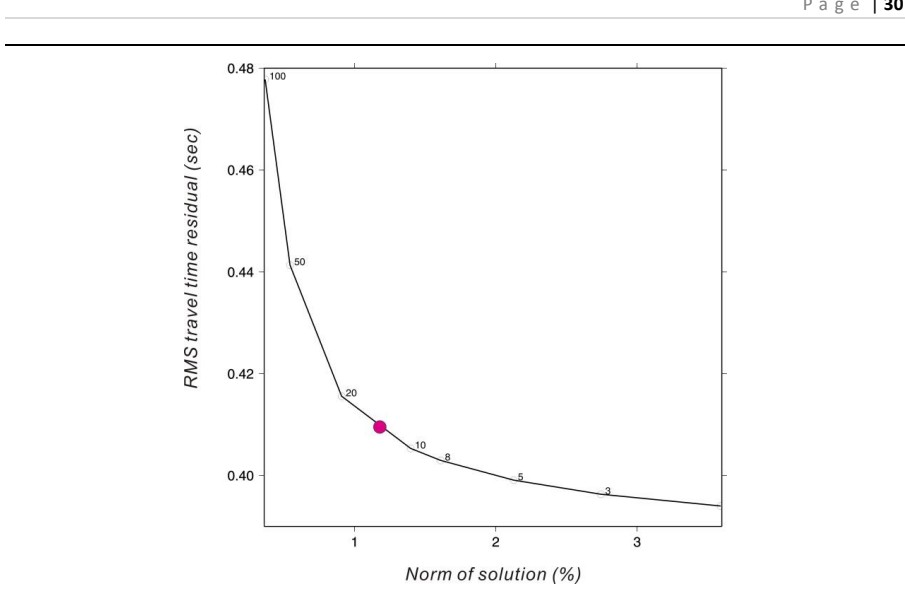


Fig. 4 The damping parameter (15) taken to invert final solution model (red circle) for
CRT and synthethic tests after a series inversion test. RMS travel time residual is about

0.41 s.



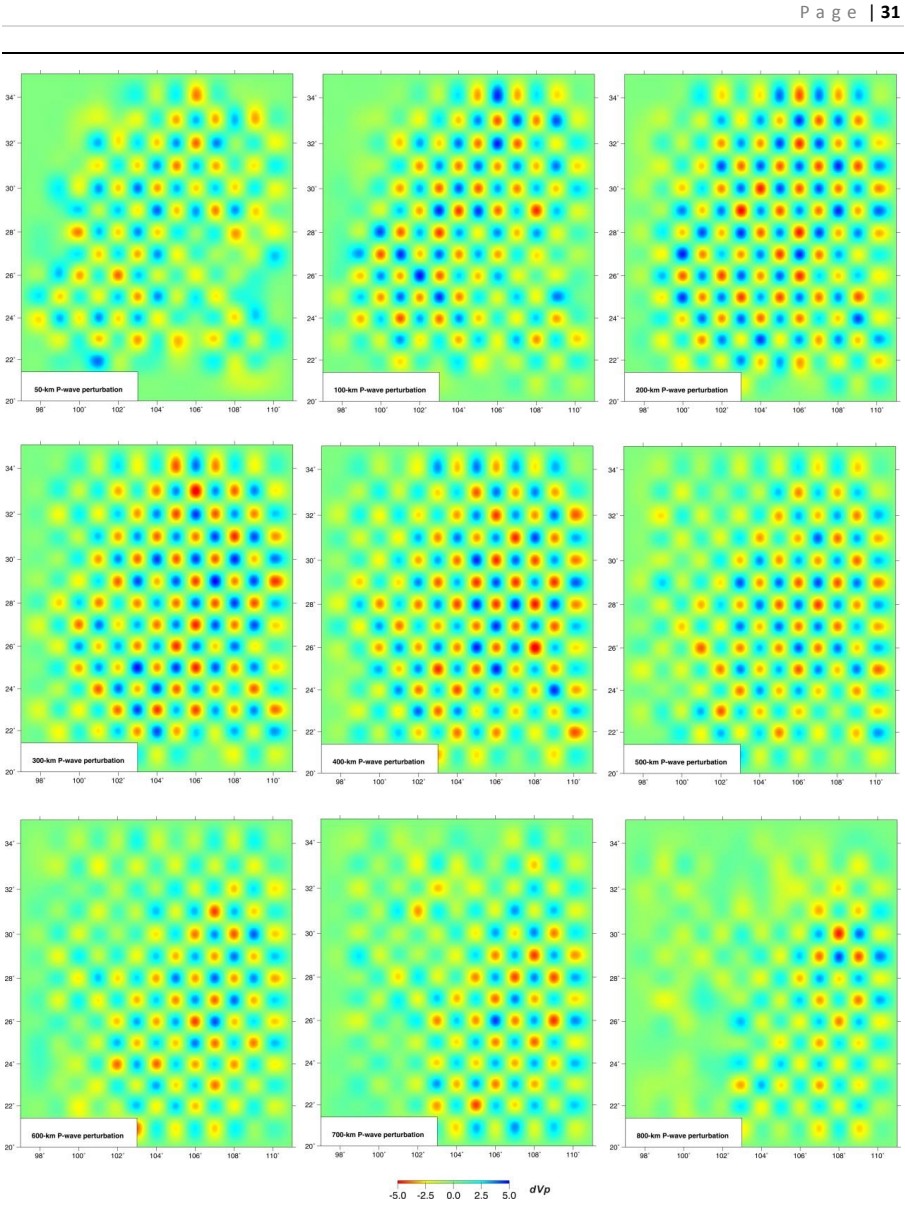


Fig.5 Checkboard resolution test at 50, 100, 200, 300, 400, 500, 600, 700 and 800 km

depth sections relative to IASP91 1D velocity model (Kennett and Engdahl, 1991). The

model was run using the same raypaths as the main inversion, with the same damping

parameter.







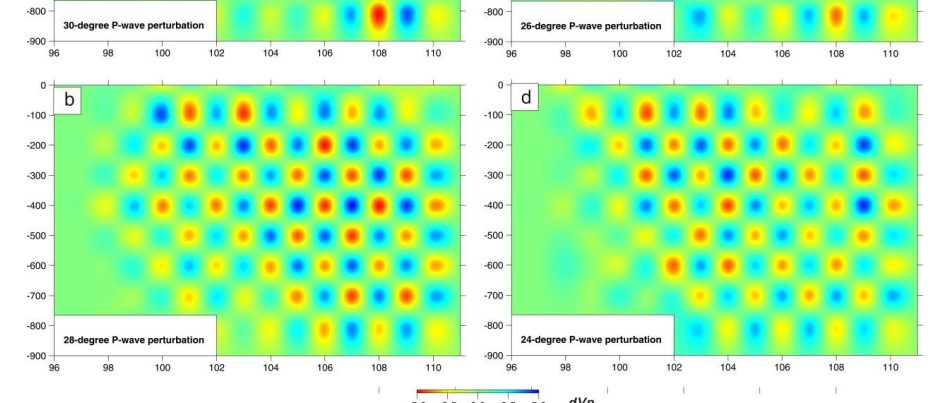


Fig. 6 Checkboard resolution test along the west-east direction profiles (a, b, c, and d is
latitude 24°N, 26°N, 28°N and 30°N direction, respectively) (see Fig. 1 for profile
location).



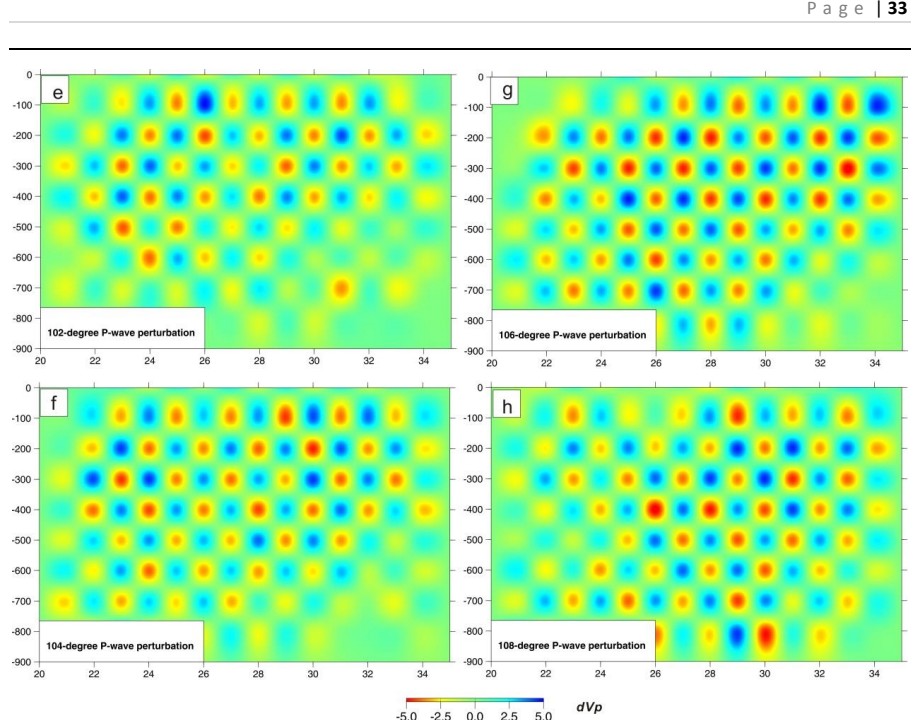


Fig. 7 Checkboard resolution test along the north-south direction profiles (e, f, g and h
are sections along longitude 102° E, 104° E, 106° E and 108° E, respectively) (see
Fig. 1 for profile location).

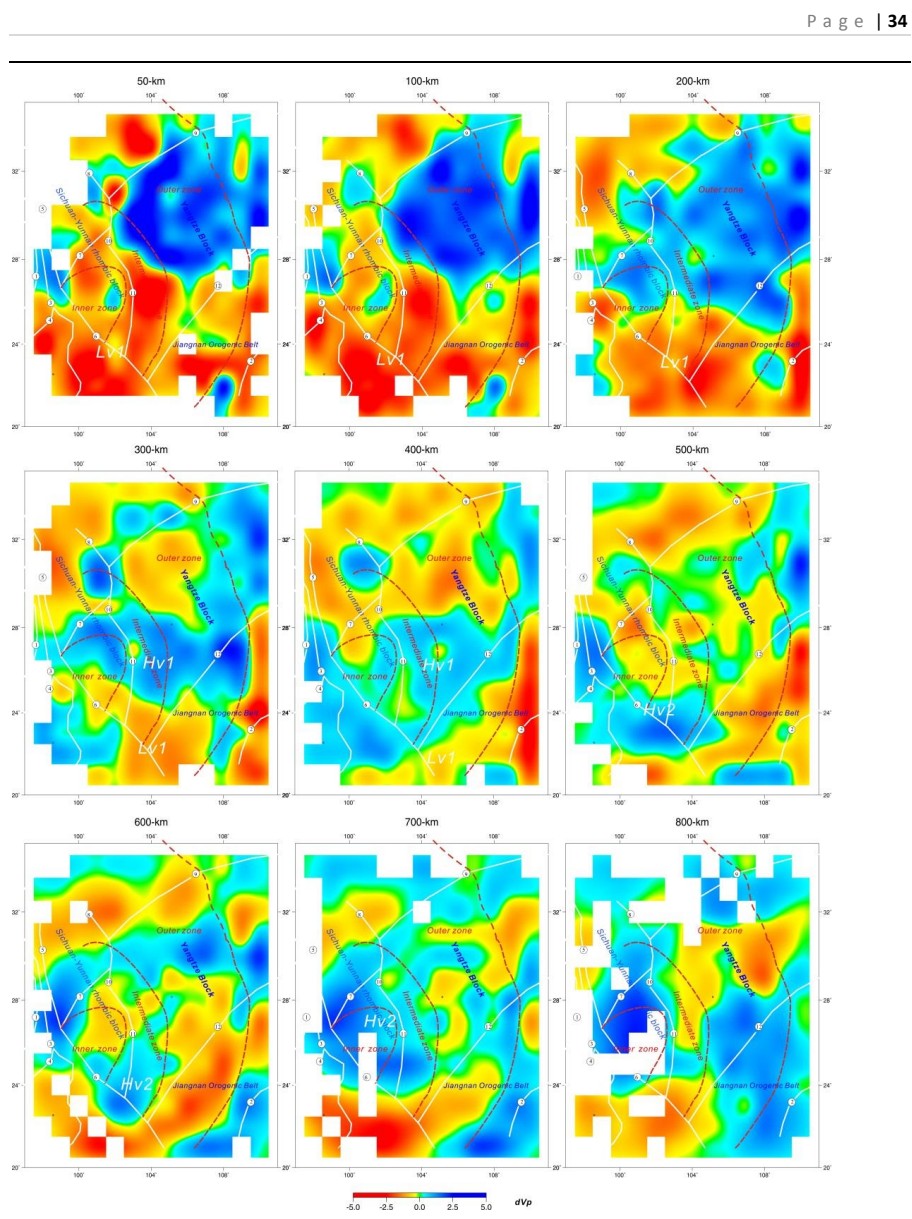


Fig. 8 P-wave velocity perturbation at 50, 100, 200, 300, 400, 500, 600, 700 and 800

km depth sections relative to IASP91 1D velocity model (Kennett and Engdahl, 1991).

Portions of the model where the recovery of the starting model in the CRT was below 10%

are not shown (see Fig. 5).



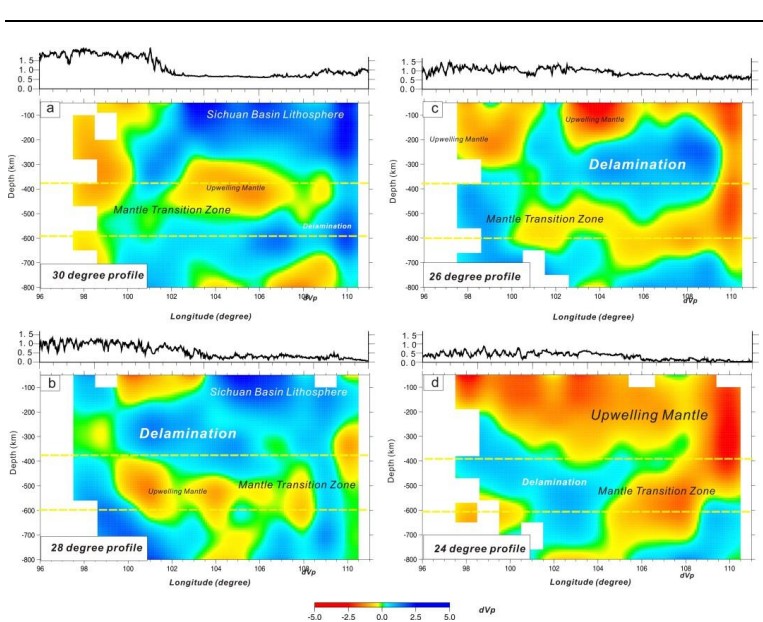



Fig. 9 P-wave velocity perturbation profiles along the west-east direction (a, b, c, and
d is latitude 24°N, 26°N, 28°N and 30°N direction, respectively) (see Fig. 1 for profile
location). Portions of the model where the recovery of the starting model in the CRT was
below 10% are not shown (see Fig. 6).



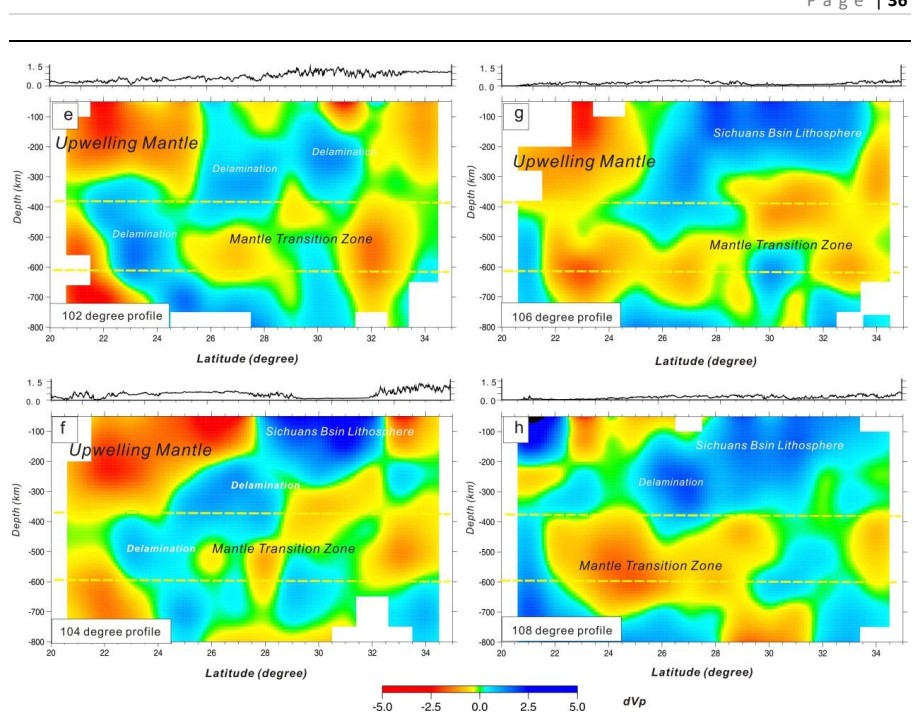


Fig. 10 P-wave perturbation profiles along the north-south direction (e, f, g and h are

sections along longitude 102° E, 104° E, 106° E and 108° E, respectively) (see Fig. 1

for profile location). Portions of the model where the recovery of the starting model in the

CRT was below 10% are not shown (see Fig. 7).

656

657

658