# Peer review of "Mantle roots of the Emeishan plume: an evaluation based on telesismic P-wave tomography"

_Solid Earth, 2017_

## Referee Comment (RC1) · G. Nolet (Referee) · 17 Apr 2017

The tomographic image of the Emeishan LIP in southwestern China, is explained by "melting of delaminated lower crustal and/or lithospheric components and associated plume-like upwelling from the mantle transition zone". In other words, a shallow origin. Such a surprising conclusion requires a scrupulous argumentation, based on verifiable predictions and hard evidence. I am afraid I do not quite see that in this paper, which presents the results of a new tomographic study with some differences with earlier images (that seems to have used more stations than this study). The argumentation is less than compelling, it does not clearly argue how today's tomographic images can give relevant information for an eruption that occurred more than 250 my ago. Laurasia

at the time was certainly not at the same location, between 200-160 my ago it moved rapidly northwards, its earlier history is not solidly documented as far as I know (see Seton et al., Earth-Science Reviews 113:212–270, 2012). Anything visible below the lithosphere at its present location is therefore irrelevant for the interpretation of the Emeishan LIP.

This is my main objection to this paper. Let me try to pose a number of other questions and give some comments.

On page 2 the authors do not clearly distinguish between the causes of seamounts and LIPS, which is confusing. Whereas seamounts are often thought to be caused by secondary or shallow, lasting plume activity, the volume erupted in the short time span observed for LIPS are probably caused by the impact of a large head from a massive (and supposedly deep) plume, not the plume tail. The volume of a LIP is of the order of $10^6$ km$^3$, very much larger than that of a seamount, their causes must thus be very different. Eimashan is characterized by a rapid, large volume ($0.25*10^6$ km$^3$) eruption, typical for a LIP. Being accessible by land-based seismic stations, it could thus be a prime candidate for testing of the plume head impact hypothesis if one can clearly link this to present day lithospheric structure. This missing link might be provided by geodynamical calculations leading to predictions for the present day, but that obviously introduces a whole new class of degrees of freedom, so it is not obvious how to interpret even the shallow tomography in terms of the causes of an ancient LIP.

The authors use the stations from the CSN, but not of the Chin array project or other temporary deployments, for reasons not discussed. The span of the array is thus comparable to that used by Huang et al. (2015), but the station density is inferior. The span is not very large, it covers an area of about 600x800 km. Not surprisingly the resolution is therefore limited to the upper mantle (fig 5), as was the result by Huang et al. (2015). There are a few small but interesting differences between the two solutions, for example at 500 km depth where He finds a strong low velocity anomaly beneath the Jiangnan orogenic belt, unlike Huang. And at 700 km Huang finds a plume-like

anomaly beneath the Yangtze block that is absent in He's analysis. But here we are at the limit of resolution, so one wonders if these differences are significant. And, as I argued before, the deeper structure at this location cannot be linked to the LIP because that event must have taken place at a much lower latitude.

Unfortunately, even the resolution test leaves questions. Were any errors added to the synthetic data? If not, the test is too optimistic and the level of 10% adopted for whitening out of the solution (figure 8-10) is probably far to low. If they did include errors in the test, what standard error was adopted for the delay times? Figure 4 seems to indicate that an error of about 0.4s is realistic.

The discussion of the mechanism for the LIP formation is not very transparent and mixes several possible scenarios. The absence of evidence for an uplift and for under-plating is used to question the plume hypothesis. This is the closest the authors come to formulating a testable prediction.

Both studies detect high velocity zones near 300 km depth. He et al. argue for a role of delamination in the Eimashan eruption. However, this leaves me again with a question about timing. If the delaminated slab has anything to do with causing the LIP 260my ago, it would by now surely have sunk deep into the mantle further south. He et al. speculate that a water-rich transition zone might stop it, but (1) the high velocities are shallower and (2) wouldn't water lower the density of the surrounding mantle and have a slab sink even faster? Slabs normally dehydrate when sinking, and surely the water cannot diffuse back so quickly into the slab that it lowers its density? And (3) it is at the wrong location.

In summary, it does not become clear why these tomographic images should be preferred to those of Huang (where they differ). And one cannot link images of the transition zone with the Eimashan.

A final, serious, remark: the number of references is very excessive (more than one hundred!), and this smells suspiciously like manipulation of the citation indices. It can

be reduced substantially. Just to give one example, the authors state (line 104) that the velocity is determined by linear interpolation among eight nodes (more accurate would be to say "trilinear interpolation"). This is such an elementary operation that it could be stated with one or even without any reference, but it is followed by no less than ten references. And each of these references is to the same person. . .

---

## Referee Comment (RC2) · N. Rawlinson (Referee) · 18 Apr 2017

In this paper, the authors use teleseismic tomography to investigate the upper mantle beneath the Emeishan LIP in southwestern China. Based on the imaging results, a number of inferences are made with regard to the mantle origin of the volcanics. I have a number of comments, both minor and major, which are listed below:

(1) Line 16-17: "Our results do not provide any conclusive evidence for upwelling mantle plume rooted in the CMB beneath the Emeishan LIP" - given that the model region extends down to 800/900 km depth and the mantle layer is ∼3000 km thick, there is no prospect of conclusive evidence for a plume rooted in the CMB, irrespective of what the results show.

[Figure]

(2) In general, the written English is ok, but there are a number of places where the sentences don't quite make sense (e.g. Lines 62-65).

(3) Lines 74-76: If the LIP formed a quarter of a billion years ago, present day mantle dynamics are unlikely to be able to help shed light on its origin.

(4) Line 91: "velocity structure or mantle dynamics" - if you apply traveltime tomography, then there is no choice but to recover velocity structure. You can only make inferences about the mantle dynamics from these results.

(5) Lines 102-103: What 1D velocity model is used, and what is the "fast raytracing technique"? I know that references are provided, but a one sentence summary would be useful.

(6) Lines 104-107: Is this a regular grid in spherical coordinates? And linear interpolation is not possible in 3-D when the velocity field is a function of 8 surrounding control points (in general, gradV will not be constant inside a cell). I think most people refer to it as pseudo-linear interpolation. This parameterization is fairly standard, so why are so many papers by Zhao cited in lieu of a definition?

(7) Lines 111-112, and Figure 2 caption: It is stated that events between 30-85 degrees angular distance are used, but the plot shows events out to $\sim$100 degrees. Azimuthal coverage is very varied too, with most events from the south and east. In order to improve coverage from other quadrants, is there any prospect of employing other global phases, such as PP, SKP, Pdiff etc?

(8) Line 115: Does capping the maximum residual mean that larger residuals are due to noise or an inability to model signal? In practice, this difference is perhaps not worth dwelling upon, but I always find it interesting when influential data (large residuals demand more significant model perturbations) are ignored.

(9) Line 118: But at least in Figure 6 (for example), the model seems to extend in depth to 900 km, not 800 km.

(10) Line 119: The contribution of the crust to the pattern of teleseismic arrival time residuals can be quite significant, particularly in regions with large changes in elevation. More information about how this correction was made and which model was used would be useful. Looking at the results (e.g. Figure 9), shallow high velocity zones tend to be associated with regions of low elevation, where one would expect thinner crust, and hence a negative contribution to the arrival time residual compared to where there is thicker crust. Hence, if the crustal correction doesn't adequately take this into account, it may result in artefacts in the upper mantle velocity structure.

(11) Lines 122-126: Why only quantify (and illustrate) the damping vs. data fit trade-off if smoothing is also applied? Also, is the tomography iterative non-linear, or just linear (rays only traced through reference model)?

(12) Paragraph starting line 128: "...tracing the actual rays through a synthetic structure..." What is meant by the "actual rays"? Are these the ray geometries from the initial or final model of the observational study? In other words, is the checkerboard test a purely linear inverse problem? At the risk of blowing my own trumpet, I suggest reading the paper:

N. Rawlinson, W. Spakman; On the use of sensitivity tests in seismic tomography. Geophys J Int 2016; 205 (2): 1221-1243. doi: 10.1093/gji/ggw084

which outlines some of the pitfalls of using a synthetic recovery test with a relatively tight pattern of uniformly-sized anomalies. I think in this case the quality of the recovery overstates what can actually be achieved with the real data. Apart from data noise, which I doubt is accounted for here, this kind of test is strongly preconditioned to produce favourable results. For instance, there is very little evidence of smearing, yet the use of teleseismic body waves generally results in some near-vertical smearing. Finally, lines 135-136 simply repeat lines 117-118 for no good reason.

(13) Lines 154-155: Given that this is a teleseismic tomography study based on relative arrival time residuals, velocity perturbations can only be discussed in a relative rather

than absolute sense, unless constraints from elsewhere are applied.

(14) Line 157-159: The recent study by Huang et al (2015) is frequently referred to, and indeed one lingering question is what the new teleseismic tomography study brings to the table that the 2015 study does not. A quick comparison of the arrays used show that Huang has a much denser station network, but in this study the array extends further north and east. The implications of these differences should be discussed somewhere. Also, Huang jointly inverts a large database of local earthquakes along with the teleseisms in order to constrain crustal structure, which ostensibly is an advance on what is done in the current study. Therefore, some discussion and perhaps justification of the current study relative to those that precede it is probably warranted.

(15) Line 197: "...with crust and (or) lithospheric delamination." - It would be more correct to say "mantle lithosphere" rather than just "lithosphere".

(16) Line 225: How does a receiver function study demonstrate convective circulation in the mantle? A few more details of this study would be useful to include.

(17) Line 248: Following from the above, how do the receiver functions identify felsic lower crust?

(18) Lines 260-264: Getting back to an earlier point I made, I would be very careful about associating what you see at 500-600 km depth with a LIP that formed a quarter of a billion years ago. I'm not saying that there cannot be an association, but it would be good to see some independent evidence e.g. from geodynamic modelling. If the delaminated lithosphere is distributed as inferred by this study, how long would it take to go from initial separation to this state, and given plate motion over time, is it located where you would expect? And when might the volcanism occur? While undertaking modelling of this type is clearly beyond the scope of this study, I still think that the readers of the journal will need a bit of convincing that what is proposed is actually possible.

(19) The Conclusions are rather brief, and would benefit from being fleshed out a bit.

Nick Rawlinson

---

## Author Comment (AC1) · 23 Apr 2017

The tomographic image of the Emeishan LIP in southwestern China, is explained by "melting of delaminated lower crustal and/or lithospheric components and associated plume-like upwelling from the mantle transition zone". In other words, a shallow origin. Such a surprising conclusion requires a scrupulous argumentation, based on verifiable predictions and hard evidence. I am afraid I do not quite see that in this paper, which presents the results of a new tomographic study with some differences with earlier images (that seems to have used more stations than this study). The argumentation is less than compelling, it does not clearly argue how today's tomographic images can give relevant information for an eruption that occurred more than 250 my ago. Laurasia at the time was certainly not at the same location, between 200-160 my ago it moved rapidly northwards, its earlier history is not solidly documented as far as I know (see Seton et al., Earth-Science Reviews 113:212–270, 2012). Anything visible below the lithosphere at its present location is therefore irrelevant for the interpretation of the Emeishan LIP.

1. Huang et al. (2015) carried out a tomographic study used 411 temporary stations within 20-33°N and 95-110°E with 45751 relative travel time residuals. Here, we extended study region northward and eastward within 20-35°N and 97-111°E so that we can cover all region of the Emeishan LIP. We used almost same amount data with Huang et al. (2015) with 42500 relative travel time residuals, moreover, our teleseismic data is high quality and recorded by 228 permanent stations. The target of this study is with a view to construct the velocity structure and investigate mantle dynamics of the upper mantle beneath the Emeishan area, and further discuss the Emeishan LIP formation. Especially, some data recorded by the temporary stations can be opened to obtain.

2. The south China block docked with the Indochina Block on the southwest in the Triassic along the Ailaoshan-Red River fault-Song Ma suture, on the west along the Longmenshan fault, and on the north with the North China Craton along the Qinling–Tongbai–Hong'an–Dabie–Sulu orogenic belt in the early Triassic (Li et al., 2002; Zhou and Zhu, 1993; Mao et al., 2013; Zheng et al., 2013). The Emeishan LIP is considered to have formed in the Permian-Triassic (Song et al., 2013; Chung and Jahn, 1995). The LIP was broken up by the Red River fault (Xiao et al., 2004) and Longmenshan fault (He et al., 2007). However, the ~260-Ma Emeishan LIP in SW China and northern Vietnam includes voluminous continental flood basalts that are believed to have formed from same upwelling mantle (Chung and Jahn, 1995; Xu et al., 2004; Zhou et al., 2006; Wang et al., 2007; Tien, 2000; Shi and Shen, 1998). Later even located at western side of the Red River fault in the early Triassic and was displaced several hundred kilometers to the southeast by Oligo-Miocene sinistral motion along the Ailao Shan-Red River fault (Ali et al., 2005), which provides a solid evidence that the Emeishan LIP was generated after the amalgamation of the south China block and Indochina Block in the early Triassic along the Ailaoshan-Red River fault-Song Ma suture. Since then or the early Triassic, no any documents demonstrated the Emeishan LIP location change. On the other hand, a receiver function study revealed a felsic lower crust in the Emeishan area, which imply there are deep process of the crustal delamination

(He et al., 2014), at same time, the MTZ show a cold domain beneath the Emeishan LIP (He et al., 2014), which imply the delamination material (or cold material) delaminated into the upper MTZ. Generally, the crustal delamination can induce the mantle upwelling (Schott and Schmeling, 1998; Elkins-Tanton and Hager, 2000; Elkins-Tanton, 2005). Finilly, it lead to a convective circulation system between the lower crust and the MTZ beneath the Emeishan area (He et al., 2014), which also imply the Emeishan LIP still located at its formation location or present location.

3. Generally, high and low velocity relics generated by subduction slab or crustal and mantle lithospheric delamination and upwelling mantle in the asthenospheric mantle can be retained for over tens of millions of years (Cook et al. 1999; Balling 2000; Svenningsen et al., 2007; Zhai et al., 2007; He et al., 2015). These low and high velocity structure can be image by tomography (D. Zhao et al., 1992, 1994; L. Zhao et al., 2016).

On page 2 the authors do not clearly distinguish between the causes of seamounts and LIPS, which is confusing. Whereas seamounts are often thought to be caused by secondary or shallow, lasting plume activity, the volume erupted in the short time span observed for LIPS are probably caused by the impact of a large head from a massive (and supposedly deep) plume, not the plume tail. The volume of a LIP is of the order of 10ˆ6 kmˆ3, very much larger than that of a seamount, their causes must thus be very different. Eimashan is characterized by a rapid, large volume (0.25*10ˆ6 kmˆ3) eruption, typical for a LIP. Being accessible by land-based seismic stations, it could thus be a prime candidate for testing of the plume head impact hypothesis if one can clearly link this to present day lithospheric structure. This missing link might be provided by geodynamical calculations leading to predictions for the present day, but that obviously introduces a whole new class of degrees of freedom, so it is not obvious how to interpret even the shallow tomography in terms of the causes of an ancient LIP.

1. In order to avoid confusing, we have replaced "It is now widely recognized that upwelling mantle plumes generate many LIPs and numerous small chains of seamounts" with "It is now widely recognized that upwelling mantle plumes generate many LIPs".

2. Generally, high and low velocity relics generated by subduction slab or crustal and mantle lithospheric delamination and upwelling mantle in the asthenospheric mantle can be retained for over tens of millions of years (Cook et al. 1999; Balling 2000; Svenningsen et al., 2007; Zhai et al., 2007; He et al., 2015). These low and high velocity structure can be image by tomography (D. Zhao et al., 1992, 1994; L. Zhao et al., 2016).

The authors use the stations from the CSN, but not of the Chin array project or other temporary deployments, for reasons not discussed. The span of the array is thus comparable to that used by Huang et al. (2015), but the station density is inferior. The span is not very large, it covers an area of about 600x800 km. Not surprisingly the resolution is therefore limited to the upper mantle (fig 5), as was the result by Huang et

al. (2015). There are a few small but interesting differences between the two solutions, for example at 500 km depth where He finds a strong low velocity anomaly beneath the Jiangnan orogenic belt, unlike Huang. And at 700 km Huang finds a plume-like anomaly beneath the Yangtze block that is absent in He's analysis. But here we are at the limit of resolution, so one wonders if these differences are significant. And, as I argued before, the deeper structure at this location cannot be linked to the LIP because that event must have taken place at a much lower latitude.

1. Although we used seismic station is less than that of Huang et al. (2015), but, the relative travel-time residuals almost is equal, moreover, our data recorded by permanent seismic station is high quality data. We consider the resolution shouldn't have any different.
2. Above, we have explained the location of the Emeishan LIP formation.

Unfortunately, even the resolution test leaves questions. Were any errors added to the synthetic data? If not, the test is too optimistic and the level of 10% adopted for whitening out of the solution (figure 8-10) is probably far to low. If they did include errors in the test, what standard error was adopted for the delay times? Figure 4 seems to indicate that an error of about 0.4s is realistic.

We haven't added errors to the synthetic data, in this version, we increase the level of 20% adopted for whitening out of the solution. We have even added 0.1 s errors to synthetic data, but, it almost hasn't any effect on the results.

The discussion of the mechanism for the LIP formation is not very transparent and mixes several possible scenarios. The absence of evidence for an uplift and for underplating is used to question the plume hypothesis. This is the closest the authors come to formulating a testable prediction.

The rise and impingement of mantle plumes on continental and oceanic lithospheric plates would lead to the formation of mafic/ultramafic lower crust (Pirajno, 2007). However, the dominantly felsic to intermediate lower crust in this area identified from receiver function analyses (He et al., 2014, 2009; S.S. Sun et al., 2012) do not favor any large-scale underplating in the Emeishan LIP area. Alternate models consider that the LIP magmatism was triggered by decompression-induced melting of upper mantle beneath zones of lithospheric extension or fractures (Uenzelmann-Neben, 2013) which does not require any upwelling mantle plume.

Both studies detect high velocity zones near 300 km depth. He et al. argue for a role of delamination in the Eimashan eruption. However, this leaves me again with a question about timing. If the delaminated slab has anything to do with causing the LIP 260my ago, it would by now surely have sunk deep into the mantle further south. He et al. speculate that a water-rich transition zone might stop it, but (1) the high velocities are shallower and

(2) wouldn't water lower the density of the surrounding mantle and have a slab sink even faster? Slabs normally dehydrate when sinking, and surely the water cannot diffuse back so quickly into the slab that it lowers its density? And (3) it is at the wrong location.

[Figure]

Fig. 9 P-wave velocity perturbation profiles along the west-east direction (a, b, c, and d is latitude 24°N, 26°N, 28°N and 30°N direction, respectively) (see Fig. 1 for profile location). Portions of the model where the recovery of the starting model in the CRT was below 20% are not shown (see Fig. 6).

1. Please Fig. 9, there is a high velocity perturbation (Hv3) in the mantle transition zone.
2. In this study, we suggest the high velocity perturbation may be associated with the large-scale crustal and lithospheric delamination induced by the assemble of the south China block and Indochina Block in the early Triassic. We don't exclude the dehydration of the delamination material, however, our results show the high velocity perturbation still exists at upper mantle, which imply Hv2 can sink into deep location.
3. Geological study and previous receiver function demonstrated the Emeishan LIP location isn't changed.

In summary, it does not become clear why these tomographic images should be preferred to those of Huang (where they differ). And one cannot link images of the transition zone with the Eimashan.

1. The target of this study is with a view to construct the velocity structure and

investigate mantle dynamics of the upper mantle beneath the Emeishan area, and further discuss the Emeishan LIP formation.

2. A large-scale lower crustal and (or) mantle lithospheric delamination or sinking (Hv3) may get arrested at the 660 km discontinuity identified by this study, where crustal and lithospheric components would be melted (Lustrino, 2005) because the MTZ is a potential water reservoir in the Earth's interior (Karato, 2011; Kuritani et al., 2011). Accumulation of delaminated crust and (or) lithosphere at the MTZ are speculated to give rise to 'second continents' on the bottom of the upper mantle (Kawai et al., 2013; Korenaga, 2004, Lustrino, 2005). The minerals in Earth's MTZ as 'water tanks' might trigger dehydration melting of vertically flowing mantle (Schmandt et al., 2014). Because of their buoyancy, crustal and (or) mantle lithospheric melts rise up as plume-like upwelling instead of being dragged down to the convecting lower mantle (Lustrino, 2005). Thus, lower crustal and (or) mentle lithospheric delamination and mantle inflow are considered to set the ideal scene for plume-like upwelling from the MTZ (He et al., 2014), which contribute to the Emeishan LIP formation.

A final, serious, remark: the number of references is very excessive (more than one hundred!), and this smells suspiciously like manipulation of the citation indices. It can be reduced substantially. Just to give one example, the authors state (line 104) that the velocity is determined by linear interpolation among eight nodes (more accurate would be to say "trilinear interpolation"). This is such an elementary operation that it could be stated with one or even without any reference, but it is followed by no less than ten references. And each of these references is to the same person: : :

Sorry, this have been revised, please see references.

---

## Author Comment (AC2) · 23 Apr 2017

(1) Line 16-17: "Our results do not provide any conclusive evidence for upwelling mantle plume rooted in the CMB beneath the Emeishan LIP" - given that the model region extends down to 800/900 km depth and the mantle layer is ~3000 km thick, there is no prospect of conclusive evidence for a plume rooted in the CMB, irrespective of what the results show.

This have been rewritten, please see below:
Our results do not provide any conclusive evidence that upwelling mantle from the lower mantle which exclude upwelling mantle plume rooted in the core-mantle boundary (CMB) beneath the Emeishan LIP.

(2) In general, the written English is ok, but there are a number of places where the sentences don't quite make sense (e.g. Lines 62-65).

This have been revised, please see below:
Previous studies suggested the Emeishan flood basalts generated by mantle plume impingement at the base of the lithosphere and caused large-scale regional up-doming prior to volcanism (Shellnutt et al., 2012; Shellnutt, 2013; Li et al., 2002; Gao et al., 1999; Liu et al., 2008) and led to a short eruption of less than 1 Ma (Song et al. 2004).

(3) Lines 74-76: If the LIP formed a quarter of a billion years ago, present day mantle dynamics are unlikely to be able to help shed light on its origin.

The tectonic framework of Emeishan LIP is characterized by the Longmenshan thrust fault in the northwest and the Ailaoshan-Red River strike slip fault in the southwest. It is possible that the assembly of Yangtze block with another crustal block in the Late Permian and Early Triassic, which is the largest tectonic event in the Emeishan area, might have led to crustal thickening and large-scale delamination of the lower crust and (or) mantle lithosphere. The delamination resulted in the mantle upwelling and generation of crustal melts that triggered plume-like upwelling, eventually, resulting in the Emeishan LIP formation.

Generally, high and low velocity relics generated by subduction slab or crustal and mantle lithospheric delamination and upwelling mantle in the asthenospheric mantle can be retained for over tens of millions of years (Cook et al. 1999; Balling 2000; Svenningsen et al., 2007; Zhai et al., 2007; He et al., 2015). These low and high velocity structure can be image by tomography (D. Zhao et al., 1992, 1994; L. Zhao et al., 2016).

(4) Line 91: "velocity structure or mantle dynamics" - if you apply traveltime tomography, then there is no choice but to recover velocity structure. You can only make inferences about the mantle dynamics from these results.

Thank you, this have been revised, please see below:
The target of this study is with a view to construct the velocity structure and investigate mantle dynamics of the upper mantle beneath the Emeishan area, and further discuss the Emeishan LIP formation.

(5) Lines 102-103: What 1D velocity model is used, and what is the "fast raytracing technique"? I know that references are provided, but a one sentence summary would be useful.

Thank you, it is revised, please below:
The location of the seismic ray crosses through the boundary of the study region was determined by a 1D (or 1D IASP91) velocity model.

(6) Lines 104-107: Is this a regular grid in spherical coordinates? And linear interpolation is not possible in 3-D when the velocity field is a function of 8 surrounding control points (in general, gradV will not be constant inside a cell). I think most people refer to it as pseudo-linear interpolation. This parameterization is fairly standard, so why are so many papers by Zhao cited in lieu of a definition?

Thank you, it has been revised:

Theoretical travel time and seismic ray paths are obtained by the fast raytracing technique (or pseudobending technique) (Um and Thurber, 1987; Zhao et al., 1992). 3D grids are employed to express the velocity perturbation values, and any point in the model space can be calculated from values of the surrounding eight nodes by pseudo-linear interpolation (Zhao et al., 1992, 1994).

(7) Lines 111-112, and Figure 2 caption: It is stated that events between 30-85 degrees angular distance are used, but the plot shows events out to ~100 degrees. Azimuthal coverage is very varied too, with most events from the south and east. In order to improve coverage from other quadrants, is there any prospect of employing other global phases, such as PP, SKP, Pdiff etc?

Epicenter distance range from 30 degree to 85 degree for each station-event pair rather than for the center of study region-event pair. In this study, we only used P-travel-time arrival, based on our assessment and comparing with other similar studies in this area, our results should be accepted.

(8) Line 115: Does capping the maximum residual mean that larger residuals are due to noise or an inability to model signal? In practice, this difference is perhaps not worth dwelling upon, but I always find it interesting when influential data (large residuals demand more significant model perturbations) are ignored.

Generally, we suggest the larger residual are due to noise or an inability signal.

(9) Line 118: But at least in Figure 6 (for example), the model seems to extend in depth to 900 km, not 800 km.

Thank you, it is revised.

(10) Line 119: The contribution of the crust to the pattern of teleseismic arrival time residuals can be quite significant, particularly in regions with large changes in elevation. More information about how this correction was made and which model was used would be useful. Looking at the results (e.g. Figure 9), shallow high velocity zones tend to be associated with regions of low elevation, where one would expect thinner crust, and hence a negative contribution to the arrival time residual compared to where there is thicker crust. Hence, if the crustal correction doesn't adequately take this into account, it may result in artefacts in the upper mantle velocity structure.

In this version, we further explain the crustal correction, please see below:

In teleseismic tomography, rays do not crisscross well in the crust and the uppermost mantle beneath the study region. Therefore, the effect of crustal heterogeneity need to be removed through correcting the relative travel-time residuals, which is called crustal correction (Zhao et al., 2006; Jiang et al., 2009, 2015). In this work, the CRUST1.0 model (Laske et al., 2012) is used to make the crustal correction to the relative travel-time residuals following the scheme of Jiang et al. (2015). Here, we are calculating the crustal correction for the upper 50 km of the earth.

(11) Lines 122-126: Why only quantify (and illustrate) the damping vs. data fit trade-off if smoothing is also applied? Also, is the tomography iterative non-linear, or just linear (rays only traced through reference model)?

In this code, teh smoothing parameter is fixed, so we only quantity the damping vs data fit trade-off.
In this study, the LSQR algorithm (Paige and Saunders, 1982) was used to solve the large and sparse system of observation equations with damping and smoothing regularizations (Zhao, 2004).

(12) Paragraph starting line 128: "...tracing the actual rays through a synthetic structure..." What is meant by the "actual rays"? Are these the ray geometries from the initial or final model of the observational study? In other words, is the checkerboard test a purely linear inverse problem? At the risk of blowing my own trumpet, I suggest reading the paper:

N. Rawlinson, W. Spakman; On the use of sensitivity tests in seismic tomography. Geophys J Int 2016; 205 (2): 1221-1243. doi: 10.1093/gji/ggw084

which outlines some of the pitfalls of using a synthetic recovery test with a relatively tight pattern of uniformly-sized anomalies. I think in this case the quality of the recovery overstates what can actually be achieved with the real data. Apart from data noise, which I doubt is accounted for here, this kind of test is strongly preconditioned to produce favourable results. For instance, there is very little evidence of smearing, yet the use of teleseismic body waves generally results in some near-vertical smearing. Finally, lines 135-136 simply repeat lines 117-118 for no good reason.

The repeat lines have been removed.

Thank you, this part have been improved, please see below:

For evaluating the resolution of the 3-D velocity structure, we carried out checkerboard resolution tests (CRTs) (Zhao et al., 1994; Zhao, 2001; Rawlinson and Spakman, 2016) and assigned positive and negative velocity perturbations of ±5% to all the 3D grid nodes. Synthetic travel times are calculated for the checkerboard model. The locations for stations/events are the same in the synthetics as in the real data. We then inverted the synthetic data with the same algorithm as that for the real data. Although the CRTs have a number of potential drawbacks (Rawlinson and Spakman, 2016), however, it bascally reflects the resolution of the tomography and become method of rotine checking. (13) Lines 154-155: Given that this is a teleseismic tomography study based on relative arrival time residuals, velocity perturbations can only be discussed in a relative rather than absolute sense, unless constraints from elsewhere are applied.

(14) Line 157-159: The recent study by Huang et al (2015) is frequently referred to, and indeed one lingering question is what the new teleseismic tomography study brings to the table that the 2015 study does not. A quick comparison of the arrays used show that Huang has a much denser station network, but in this study the array extends further north and east. The implications of these differences should be discussed somewhere. Also, Huang jointly inverts a large database of local earthquakes along with the teleseisms in order to constrain crustal structure, which ostensibly is an advance on what is done in the current study. Therefore, some discussion and perhaps justification of the current study relative to those that precede it is probably warranted.

Thank you, in this version, we have improved these, please see below:

Huang et al. (2015) carried out a tomographic study using 411 temporary stations within 20-33° N and 95-110° E and obtained the velocity structure of the crust and upper mantle in Chuandian area. Here, we carry out an extended study in the region northward and eastward within 20-35° N and 97-111° E so as to cover all the regions of the

Emeishan LIP. Although we used almost same amount of data as that of Huang et al. (2015), our teleseismic data are of higher quality and were recorded by 228 permanent stations. Our target is to construct the velocity structure and investigate the mantle dynamics beneath the Emeishan area, based on which we evaluate the geodynamics of Emeishan LIP formation.

(15) Line 197: "...with crust and (or) lithospheric delamination." - It would be more correct to say "mantle lithosphere" rather than just "lithosphere".

Thank you, this has been revised.

(16) Line 225: How does a receiver function study demonstrate convective circulation in the mantle? A few more details of this study would be useful to include.

Thank you, we have improved this part, please see below:
A receiver function study revealed a felsic lower crust in the Emeishan area, suggesting crustal delamination (He et al., 2014). Simultaneously, the MTZ beneath this region shows a cold domain (He et al., 2014), which might suggest that the delaminated cold material dropped down into the upper MTZ. Generally, crustal delamination can induce mantle upwelling (Elkins-Tanton and Hager, 2000; Elkins-Tanton, 2005), which might have eventually resulted in a convective circulation system between the lower crust and the MTZ beneath the Emeishan area (He et al., 2014). This further confirms the present location of the Emeishan LIP.

(17) Line 248: Following from the above, how do the receiver functions identify felsic lower crust?

The Vp/Vs ratio can be obtained by H-K stacking of receiver function technique, generally, the crustal delamination resulted in felsic lower crust with low Vp/Vs ratio (<1.75).

(18) Lines 260-264: Getting back to an earlier point I made, I would be very careful about associating what you see at 500-600 km depth with a LIP that formed a quarter of a billion years ago. I'm not saying that there cannot be an association, but it would be good to see some independent evidence e.g. from geodynamic modelling. If the delaminated lithosphere is distributed as inferred by this study, how long would it take to go from initial separation to this state, and given plate motion over time, is it located where you would expect? And when might the volcanism occur? While undertaking modelling of this type is clearly beyond the scope of this study, I still think that the readers of the journal will need a bit of convincing that what is proposed is actually possible.

1.  Previous receiver function have demonstrated the Emeishan LIP formation may be associated with the delamination, since then, there aren't any evidence demonstrated

the location of the Emeishan LIP change (He et al., 2014).
2. The vestige of the delamination (High velocity perturbation) and mantle upwelling (low velocity perturbation) can be retained over tens of millions years. (Cook et al. 1999; Balling 2000; Svenningsen et al., 2007; Zhai et al., 2007; He et al., 2015). These low and high velocity structure can be image by tomography (D. Zhao et al., 1992, 1994; L. Zhao et al., 2016).

(19) The Conclusions are rather brief, and would benefit from being fleshed out a bit.you

Thank you, this have been revised, please see below.

The tectonic framework of Emeishan LIP is characterized by the Longmenshan thrust fault in the northwest and the Ailaoshan-Red River strike slip fault in the southwest. It is possible that the assembly of Yangtze block with another crustal block in the Late Permian and Early Triassic, which is the largest tectonic event in the Emeishan area, leading to crustal thickening and large-scale delamination of the lower crust and (or) mantle lithosphere. The delamination resulted in the mantle upwelling and generation of crustal melts that triggered plume-like upwelling, eventually, resulting in the Emeishan LIP formation.

Our results show that there are no low velocity perturbation rooted in the lower mantle beneath the Emeishan LIP, suggesting the absence of any vestiges of a mantle plume rising from the CMB beneath the Emeishan area.

Our results also further confirm the model of vetical convective circulation in the mantle as the major trigger for the Emeishan LIP.